# Microstructure and Wear Resistance of Ni–WC–TiC Alloy Coating Fabricated by Laser

**Yu Liu [1,2,\*], Zeyu Li [1], Guohui Li [2], Fengming Du [3,\*] and Miao Yu [1]**

1   School of Mechanical Engineering, Northeast Electric Power University, Jilin 132012, China
2   International Shipping Research Institute, Gongqing Institute of Science and Technology, Gongqing 332020, China
3   Marine Engineering College, Dalian Maritime University, Dalian 116026, China
\*   Correspondence: yuliu@neepu.edu.cn (Y.L.); dfm@dlmu.edu.cn (F.D.); Tel./Fax: +86-043264807327 (Y.L.)

**Abstract:** In this study, a Ni–WC–TiC alloy coating was fabricated by laser to improve the wear resistance and service life of Cr12MoV die steel. The microstructures and phases of the coating were analyzed by a scanning electron microscope (SEM), an energy dispersive spectrometer (EDS), and X-ray diffraction (XRD). The properties of the coating were tested by a hardness and friction wear tester. The results show that the coating has a good metallurgical bond with the substrate. The microstructures from top to bottom are mainly equiaxed crystal, columnar dendrite, and cellular dendrite. Combined with the physical phase analysis and elemental distribution of the coating, there are some phases, such as $\gamma{\sim}$(Fe, Ni), $Cr_{23}C_6$, WC, TiC, $Fe_3W_3C$, and $Cr_2Ti$. Compared with the Cr12MoV steel substrate, the Ni–WC–TiC alloy coating has good properties of hardness and wear resistance. In the coating, the background region of the grains is $\gamma{\sim}$(Fe, Ni). From the EDS results, it can be seen that there are some rod-like particles, $Cr_{23}C_6$, which are uniformly distributed on the top of the coating. Some W and Ti carbides form in grains. The addition of TiC particles improves the WC particles refinement. The highest hardness of the coating is 770.7 $HV_{0.5}$, which is approximately 3.3 times higher than that of the substrate. The wear volume is 0.26 $mm^3$, or approximately 8.6% of the substrate, which is contributed to the reinforced phases and finer microstructure of the coating. The wear volumes of the Cr12MoV substrate are 1.8 and 4.5 $mm^3$ at 20 and 60 min, respectively. While the wear volumes of the Ni–WC–TiC coating are 0.2 and 0.7 $mm^3$ at 20 and 60 min, respectively. The increased amplitude of the coating's wear volume is smaller than that of the substrate. The results show that this Ni–WC–TiC alloy coating is helpful for improving the properties and service life of Cr12MoV die steel.

**Keywords:** laser cladding; Cr12MoV; Ni–WC–TiC; coating; hardness; wear resistance

## 1. Introduction

Mold is widely used in industrial production due to its advantages of good quality, high efficiency, and low cost. The Cr12MoV die steel has excellent hardenability, hardness, and flexural strength, so it is often used as a material for manufacturing molds [1,2]. The material has good thermal processing properties due to the presence of Mo and V alloying elements in Cr12MoV steel [3,4]. Therefore, it is widely made into some molds with large cross-sections and complex shapes [5,6]. However, the working conditions for the mold are harsh. In addition to the impact of force and heat, the working surface of the mold is also subjected to intense friction with the blank, which often leads to mold breakage and failure [7,8]. The requirements for mold materials are becoming more stringent, which makes the defects of Cr12MoV mold steel increasingly prominent, especially in the current situation of rapid industrial development. Therefore, it is necessary to improve the wear resistance of Cr12MoV steel.

Surface modification is one of the effective methods for improving the service life of molds [9,10]. At present, surface properties are generally improved in industry by

carburizing [11], ion implantation [12], and thermal spray strengthening [13]. However, the above surface modification methods have disadvantages, such as high operational difficulty, high cost, and easy peeling of the reinforcing layer. Compared with traditional surface modification technology, laser cladding technology, which is a laser-based additive manufacturing process with a more complex process, has the advantages of low deformation, high bonding strength, and dense organization [14–16]. It can be used not only to repair the surface of critical parts but also to produce parts of specific shapes and sizes by multi-channel and multi-layer methods [17]. Therefore, laser cladding technology is widely used in the industrial field as a new surface strengthening technology with no environmental pollution and a high degree of automation [18].

The cladding materials are usually Ni-based [19], Fe-based [20], and Co-based [21] alloy powders. Compared with the other two powders, Ni-based alloy powder has the advantages of a low melting temperature and good self-melting property [22–25]. Therefore, Ni-based alloy powder is widely used to fabricate coatings. Farzaneh et al. [26] performed a detailed defect characterization of (Ti-6Al-4V)-Ni-(Ti-6Al-4V) sandwich structures and showed that by varying the process parameters of Laser Engineered Net Shaping (LENS), it is possible to produce sandwich structures suitable for various applications. However, the hardness and wear resistance are not very good due to the lack of reinforcing phases. The ceramic particle WC is a good material to enhance the wear resistance, which is usually added into the laser cladding powder [27,28]. The fabrication of WC particle-reinforced Ni-based alloy coatings by laser cladding is widely used in actual production due to its excellent wear resistance [29–31]. Although the proper amount of WC is helpful to improve the coating's properties, an excessive amount of WC particles is prone to cracks. In order to reduce the cracks, the TiC is usually added to fabricate coatings without cracks [32,33]. In past studies, most of them prepared Ni-based alloy coatings on its surface and studied the effect of laser power on the properties of Ni-based alloy coatings. However, there are few studies about the preparation of coating with WC and TiC ceramic particles by Ni-based alloy powder on the Cr12MoV die steel. Therefore, a mixed powder of Ni–WC–TiC is proposed to fabricate a coating on the substrate of Cr12MoV die steel.

In this paper, the $CO_2$ laser was used to fabricate a Ni–WC–TiC ceramic metal-based coating on the substrate of Cr12MoV die steel. Then the morphology, microstructure, hardness, and wear resistance of the coating were studied. Analyses were carried out using a metallographic microscope, a friction and wear tester, a Vickers hardness tester, a scanning electron microscope (SEM) equipped with energy dispersive spectroscopy (EDS), and X-ray diffraction (XRD). The results show that this Ni–WC–TiC alloy coating is helpful for improving the properties and service life of Cr12MoV die steel. It is hoped that our research can reduce the production loss, save on production costs, and provide a theoretical basis for the surface modification work of Cr12MoV mold steel.

## 2. Experiment and Methods

### 2.1. Materials

The Cr12MoV steel die was used as the substrate. The main chemical composition of Cr12MoV steel was Cr 11.5~13%, Mo 0.7~1.2%, and Fe balance as shown in Table 1. The substrate's length, width, and height were 50, 30, and 15 mm, respectively. The Ni-based alloy powder was used as cladding material. The main chemical composition of Ni-based alloy powder was Cr 16.5%, Fe 7%, and Ni balance, which is included in Table 2.

**Table 1.** Elemental composition (wt.%) of Cr12MoV.

| Cr | C | V | Mo | Si | Mn | Ni | S | P | Fe |
|---|---|---|---|---|---|---|---|---|---|
| 11.5~13 | 1.4~1.6 | ≤1.0 | 0.7~1.2 | ≤0.6 | ≤0.6 | ≤0.25 | ≤0.03 | ≤0.03 | Bal. |

**Table 2.** Elemental composition (wt.%) of Ni-based alloy powder.

| Cr | Fe | Si | C | B | Ni |
|---|---|---|---|---|---|
| 16.5 | 7.0 | 4.0 | 0.8 | 4.0 | Bal. |

The Ni-based alloy powder size was 200–300 meshes, and its morphology is shown in Figure 1. The shape of the powder was mainly circular. The maximum diameter of Ni-based alloy particles was about 52 μm. There were also some small particles with several microns. There were a few stripe and block particles, which were conducive to the flow of the liquid phase during the melting process. The ceramic particles were added to the Ni-based alloy, mixed, and ground in a mortar for 30 min. Then the Ni–WC–TiC powder was made from 60% Ni-based alloy, 30% WC, and 10% TiC.

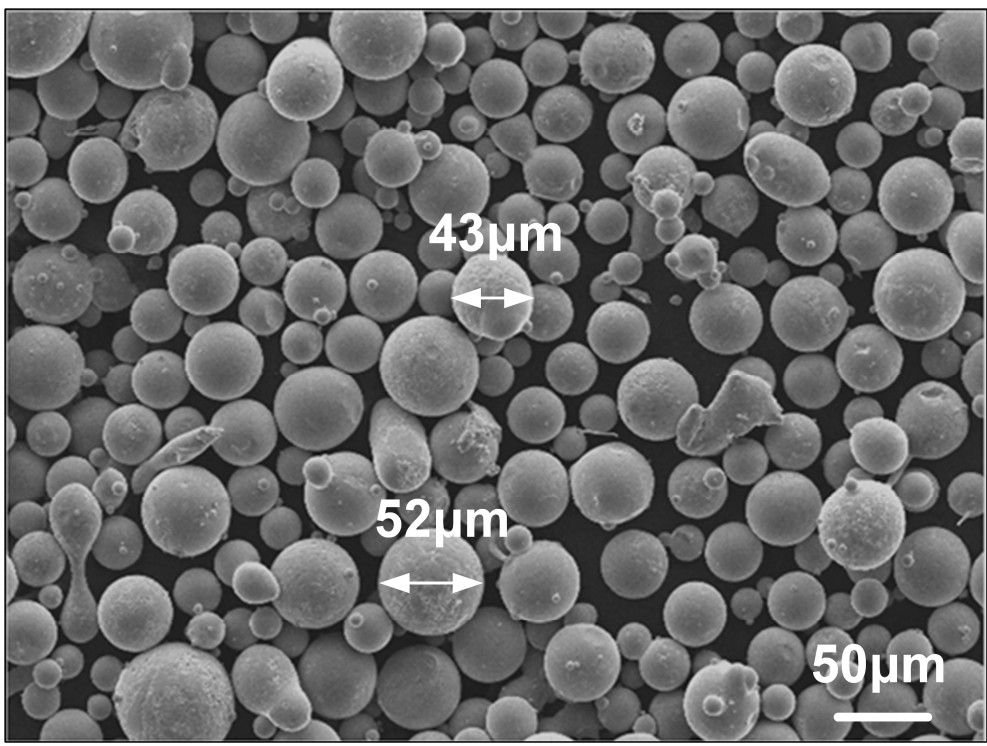

**Figure 1.** SEM micrograph of Ni-based alloy powder.

*2.2. Experimental Details*

The substrate was first sandpapered, then cleaned with acetone and dried in a drying dish for two hours to eliminate the oxide film on the surface. The Cr12MoV die steel was then rinsed with an absolute alcohol solution for 30 min in an ultrasonic machine. In this experiment, DL-HL-T2000 $CO_2$ laser cladding equipment was used for the laser cladding process. The thickness of the prefabricated Ni–WC–TiC powder is 1 mm. The laser cladding process parameters are shown in Table 3. The selected laser power was 1.65 kW, and the scanning speed was 100 mm/min. The spot diameter was 3 mm. The overlapping ratio was 30%.

**Table 3.** Laser cladding process parameters.

| Parameters | Laser Power (kW) | Scan Speed (mm/min) | Spot Diameter (mm) | Overlapping Ratio |
|---|---|---|---|---|
| Value | 1.65 | 100 | 3 | 30% |

Figure 2 depicts the schematic diagram of the laser cladding process. The energy from the laser melted the Ni-based alloy powder, which made a liquid molten pool. A

coating with a combination of Ni-based alloy and Cr12MoV substrate was formed. Then this coating of substrate was cut into four specimens along the cladding direction using a wire cutter. The length, wide, and height were all 10 mm. The coating was then polished on a polishing machine until the surface was free of scratches. Aqua regia (HCl: $HNO_3$ = 3:1) was used as a corrosive agent to corrode the cross section of the coating for 10~25 s. When light yellow bubbles appeared on the surface of the coating, it was immediately washed with absolute ethanol and blown dry.

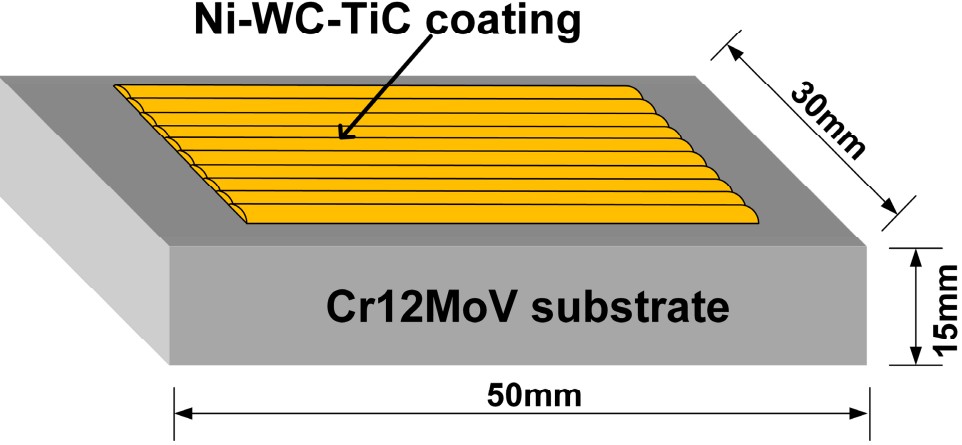

**Figure 2.** Schematic diagram of laser cladding process.

The copper target X-ray diffractometer (XRD, TD-3500X) was used to determine the phase of the coating, operating at 40 kV and 40 mA in a diffraction range of 20°~80° and a scanning speed of 4.08°/min. The cross-sectional morphology, microstructure, and composition of the coating were observed by scanning electron microscopy (SEM, ZEISS300). The elemental distribution of the Ni–WC–TiC coating was analyzed by EDS. The Vickers hardness tester (HXD-1000TMC/LCD) was used to measure the hardness of the coating with a test load of 500 gf for 10 s. The hardness was measured from the top of the Ni–WC–TiC coating to the Cr12MoV substrate, and each point was repeated three times at the same height. The reciprocating fatigue and wear tester (MGW-02) was used to test the tribological properties of the Ni–WC–TiC coating. The parameters are included in Table 4. The Φ6.5 mm GCr15 grinding ball was selected as the friction pair. The wear time was 30 min. The frequency was 2 Hz. The load was 3 N.

**Table 4.** Experimental parameters of friction and wear.

| Parameters | Wear Time (min) | Frequency (Hz) | Load (N) |
|---|---|---|---|
| Value | 30 | 2 | 3 |

*2.3. Surface Morphology*

The schematic diagram of the laser cladding fabrication process is shown in Figure 3. The parameters were selected from Table 3. The Ni–WC–TiC alloy powder was melted by a high-energy laser. The multi-pass coating was made along with the repeated movement. The fabricated Ni–WC–TiC coating was shown in Figure 3. It can be seen that the surface morphology is smooth and uniform.

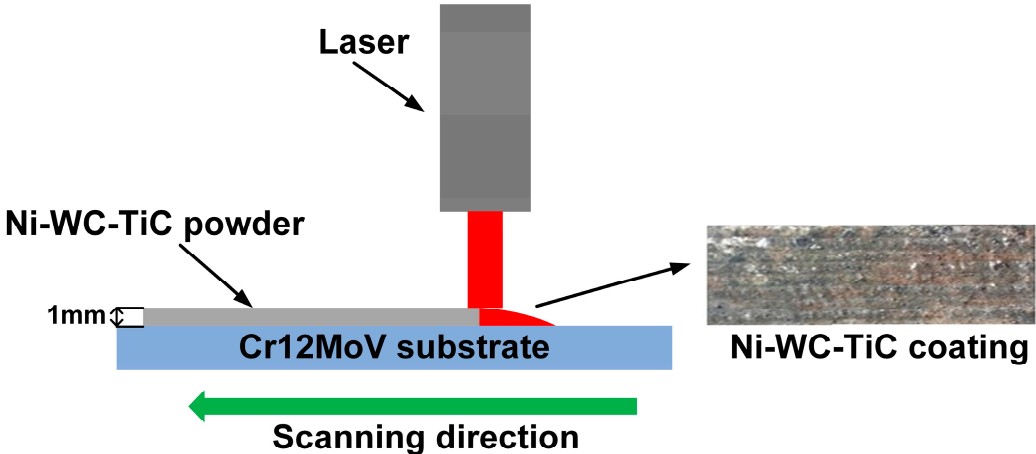

**Figure 3.** Schematic diagram of laser cladding fabrication process.

## 3. Results and Discussion

### 3.1. Phase Composition

Figure 4 shows the X-ray diffraction pattern of Ni–WC–TiC coatings. It can be found that the Ni–WC–TiC coating is mainly composed of $\gamma \sim$(Fe, Ni), $Cr_2Ti$, $Cr_{23}C_6$, WC, TiC, and $Fe_3W_3C$. Others were decomposed and formed the $Cr_2Ti$ and $Fe_3W_3C$ phases. During the formation of the coating, when the temperature of the liquid pool decreased, the $\gamma \sim$(Fe, Ni) phase with a higher diffraction peak appeared, which was detected at about 43° and 50°. The WC phase was detected at about 50° and 74°, and the TiC phase was detected at 41°, indicating that some WC and TiC were still present in the coating after the laser cladding process.

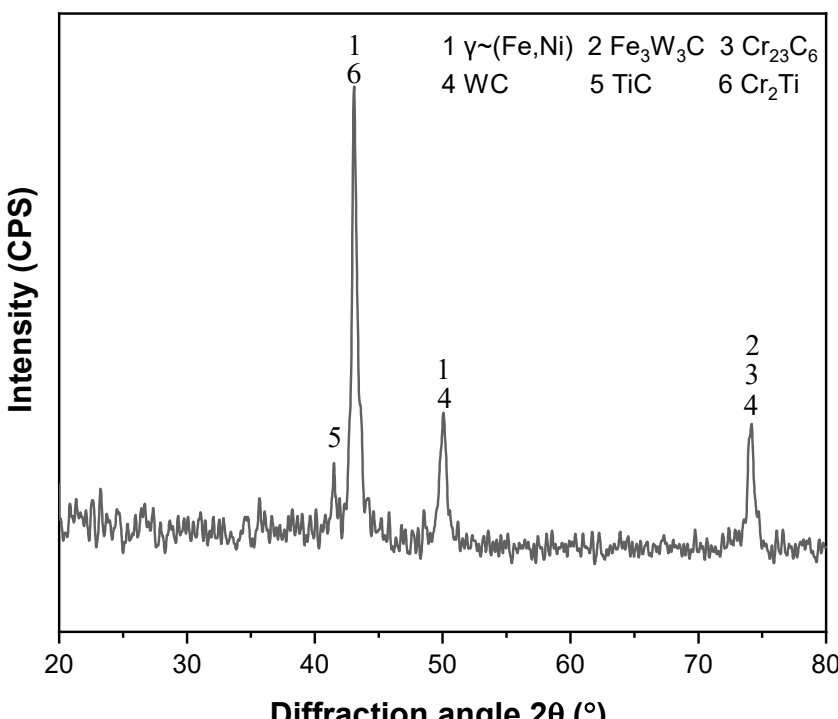

**Figure 4.** X-ray diffraction pattern of the Ni–WC–TiC coating.

Under the action of the high-energy laser beam, the Cr12MoV substrate and Ni–WC–TiC powder underwent some complex chemical and physical changes in the liquid molten pool, resulting in the formation of different phases [34]. The reaction equation is as follows:

$$2Cr + Ti \rightarrow Cr_2Ti \tag{1}$$

$$Cr + C \rightarrow Cr_{23}C_6 \tag{2}$$

$$L + WC + W \rightarrow Fe_3W_3C \tag{3}$$

where L represents the high-temperature solvent of W, C, and Fe.

Therefore, the addition of WC and TiC not only brings some ceramic particles into a Ni-based coating but also makes some new phases, such as $Cr_2Ti$ and $Fe_3W_3C$. These new phases are helpful to improve the mechanical properties of the coating.

### 3.2. Microstructure Analysis

Figure 5 shows the microstructure of the coating's overall and local regions. From top to bottom, there are a coating, bonding region, and substrate, as shown in Figure 5a. In Figure 5b, it shows the morphology of zone A, which is mainly dominated by equiaxed crystals at the top of the coating. As can be seen in Figure 5b, the size of the equiaxed crystals at zone A is about 3~5 μm. Because zone A has a larger cooling rate due to heat transfer between the molten pool and air. Then the composition overcooling phenomenon appears at the zone of liquid and solid phases, which leads to spontaneous crystallization. Therefore, a lot of equiaxed crystals form and grow. In Figure 5c, it shows the morphology of zone B, which is mostly made of columnar and cellular dendrites in the middle of the coating. As can be seen in Figure 5c, the size of the cellular dendrites in zone B is about 5~9 μm. From Figure 5c,d, it can be seen that the size of the columnar dendrites is larger, which is about 10~20 μm. The cooling rate is smaller than that of the coating's top. Therefore, the crystal has enough time to grow and become columnar dendrites and cellular dendrites. In Figure 5d, it can be seen that the bonding region appears to contain some large planar crystals. Because the temperature gradient G is larger at the bonding region while the growth rate R is smaller. The large value of G/R promotes the crystals to become a lot of planar and columnar crystals.

In order to investigate the Ni-WC-Ti coating, the points D in Figure 5c and E in Figure 5d were selected, and the elemental composition of those two points was obtained by EDS. The elemental composition results are included in Table 5. At point D, the mass fractions of Fe and Ni elements inside the crystal are 47.14% and 20.32%, respectively. The main composition of this region is γ~(Fe, Ni) solid solution, which is in accordance with the XRD pattern. The mass fractions of Cr, W, and C elements are 9.49%, 8.86%, and 11.46%, respectively, indicating that some Cr and W carbides are present in this region. At point E, the mass fractions of the W and Ti elements are 26.34% and 21.94%, respectively. This indicates that the gray-white particles in the coating are Ti and W carbides, which is consistent with the literature [35]. These gray-white particles are uniformly distributed in the grain boundary caused by the addition of WC and TiC, which are helpful to improve the coating's properties. It is worth noting that no large WC particles were found in the coating because some WC particles are decomposed under the laser's high energy. From the previous research, it can be known that the TiC particles are also helpful in making the WC particles small. Therefore, the addition of TiC particles improves the WC particles refinement.

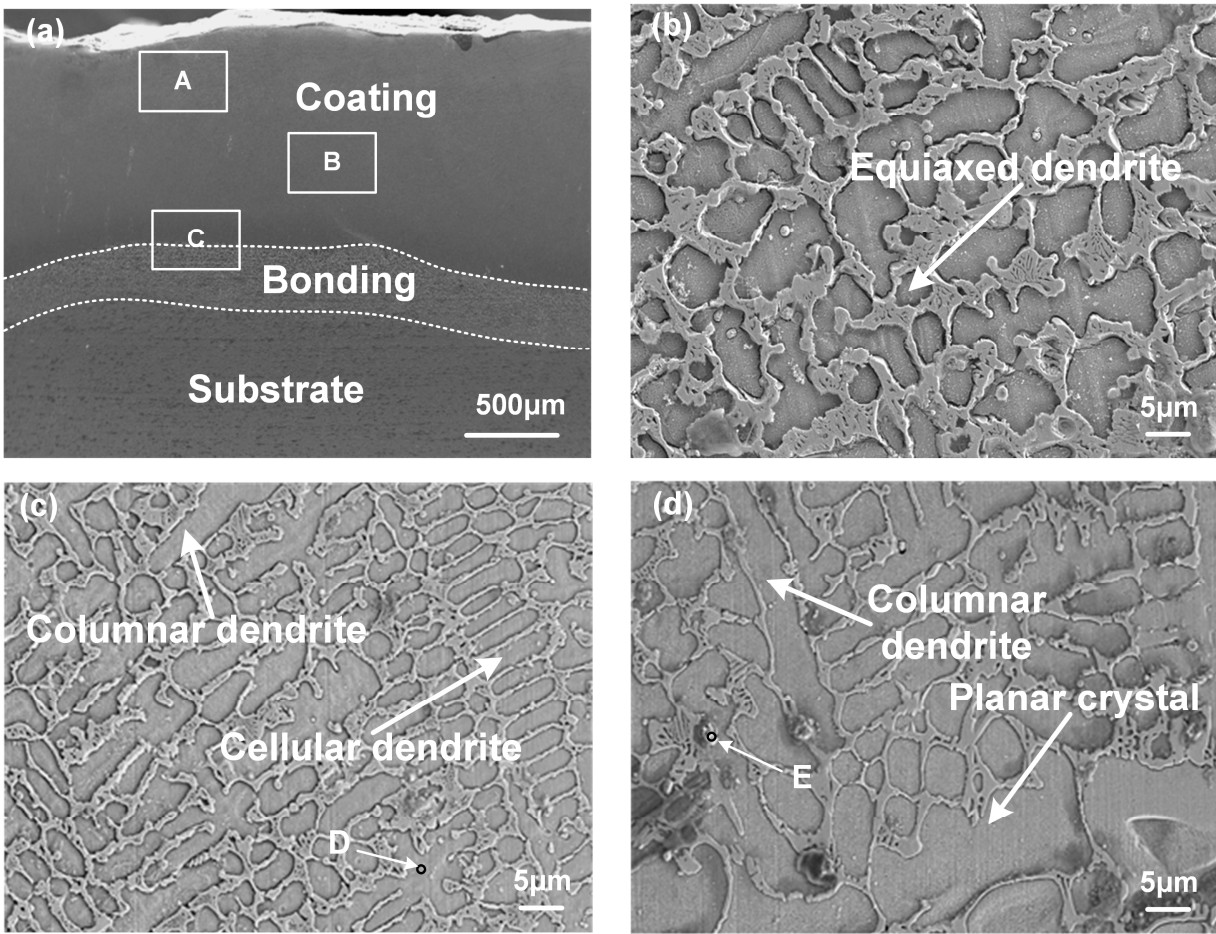

**Figure 5.** SEM images of coating; (**a**) overall morphology; (**b**) zone A; (**c**) zone B; and (**d**) zone C.

**Table 5.** Elemental composition (wt.%) of points D and E.

| Point | Fe | Ni | W | C | Ti | Cr | Si |
|---|---|---|---|---|---|---|---|
| D | 47.14 | 20.32 | 8.86 | 11.46 | 1.19 | 9.49 | 1.54 |
| E | 20.97 | 12.19 | 21.12 | 14.81 | 23.75 | 6.54 | 0.62 |

The SEM image and elemental distribution at the top of the coating are shown in Figure 6. Figure 6a shows that some relatively coarse rod-like particles are present in the coating. In Figure 6b, it can be seen that the elemental distribution is mainly dominated by the elements Fe, Ni, and Cr, supplemented by the elements W and Ti. As shown in Figure 6c,d, these rod-like structures are rich in Cr and C, which are Cr carbides. Combined with the X-ray diffraction pattern of the coating, it can be concluded that these rod-like particles are $Cr_{23}C_6$, and the background region of the grains is $\gamma\sim(Fe, Ni)$. These rod-like particles are uniformly distributed on the top of the coating as solid solutions. This is helpful for obtaining a Ni-based ceramic coating with good properties, such as hardness. This is one of the reasons for the high hardness.

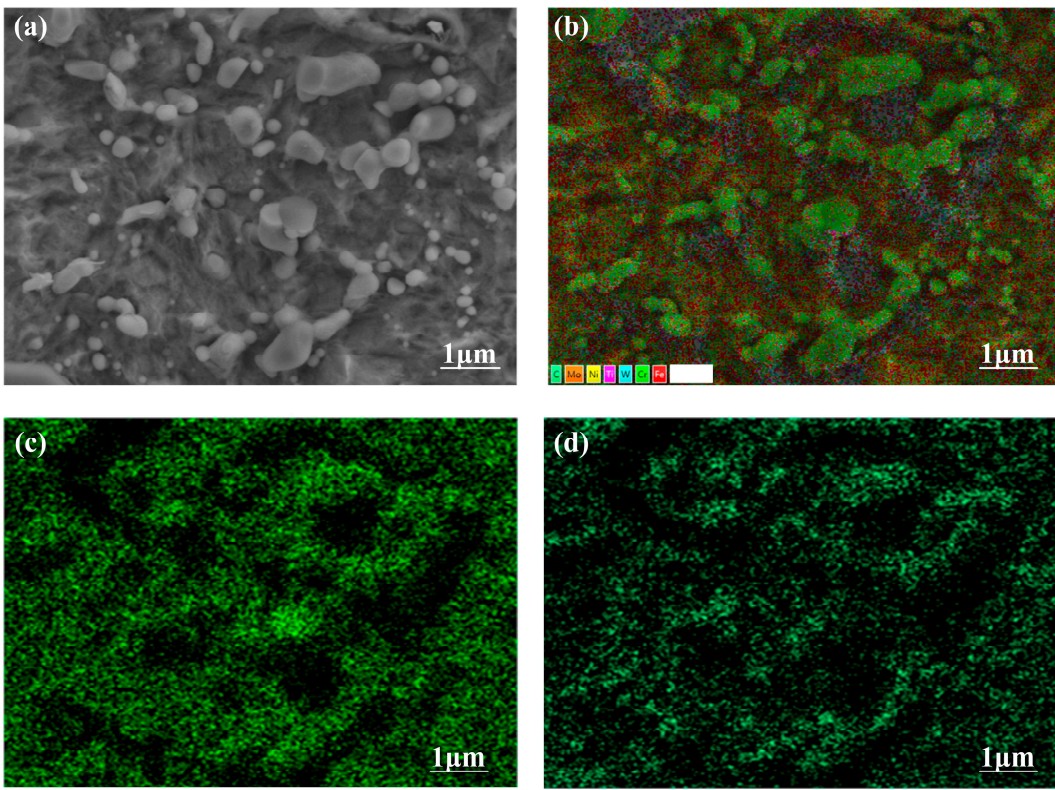

**Figure 6.** SEM image (**a**) and distribution of all elements (**b**); Cr (**c**) and C (**d**) at the top of coating.

Figure 7 shows the SEM image and elemental distribution of the grain in the Ni–WC–TiC coating. In Figure 7a, there is a SEM image of the whole region, which has a gray-white particle, grain boundary, and grain background. Figure 7b–g is the elemental distribution of W, Ti, Fe, Ni, Cr, and C. In Figure 7b,c, it can be seen that the main elements in the gray-white particle are W and Ti. There are some C elements. So these particles are W and Ti carbides. Figure 7d,e shows that the main elements of the grain background are Fe and Ni. Combined with the X-ray diffraction pattern, this is the γ~(Fe, Ni) solid solution. Figure 7f,g shows that the grain boundaries are enriched in Cr and C elements. The grain boundaries are some Cr carbides, which are helpful to increase the strength of the crystal. The TiC and WC in the powder were decomposed under the action of a high-energy laser beam. As new nuclei in the nucleation process, they play a role in increasing the nucleation rate and promoting grain refinement in the remelting process.

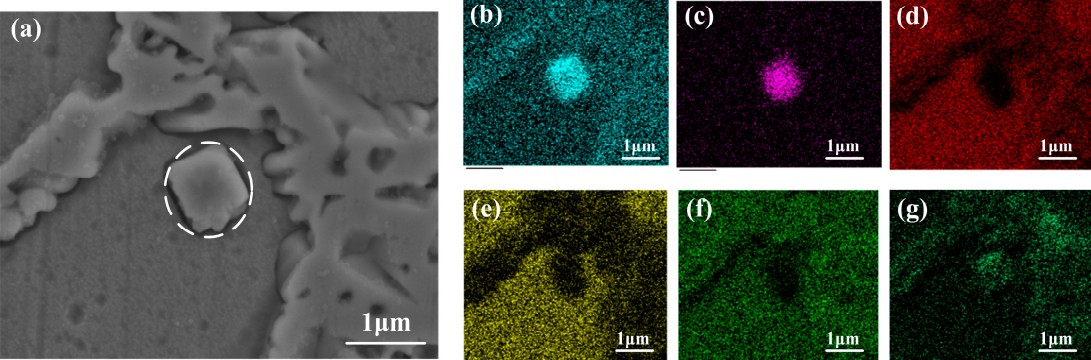

**Figure 7.** SEM image (**a**) and elemental distribution of W (**b**), Ti (**c**), Fe (**d**), Ni (**e**), Cr (**f**), and C (**g**) of the gray-white particle.

In order to investigate the elemental distribution of the bonding region, Figure 8 depicts a SEM image and the distribution of Ni, W, and Ti elements in the bonding region. Figure 8a shows the microscopic morphology of the bonding region of the coating. It can be seen that there is no cracking or porosity. The different elements have mutual penetration. Therefore, there is a good metallurgical bond between the Ni–WC–TiC coating and the substrate. Figure 8b,c shows the uniform distribution of Ni and W elements in the coating. Figure 8d shows that the Ti element is distributed beyond the coating and extends into the bonding region, indicating that the TiC is fully decomposed. Overall, the elements Ni, W, and Ti are mainly distributed in the coating and show a decreasing trend from the coating to the bonding region. It is proven that the added WC and TiC are concentrated in the coating. The microstructure and distribution of elements in the coating are helpful in fabricating a compact coating with good properties on a Cr12MoV die steel substrate, such as hardness and wear resistance.

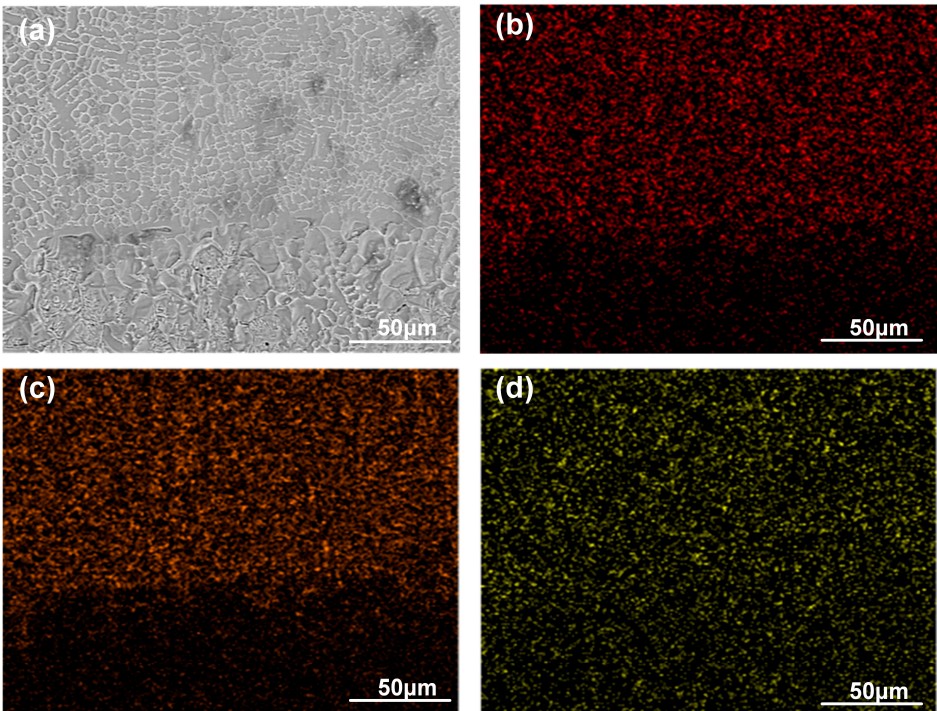

**Figure 8.** SEM image (**a**) and distribution of Ni (**b**), W (**c**), and Ti (**d**) elements at the bonding region.

*3.3. Hardness*

In order to investigate the coating's property, the hardness of the coating's cross-section perpendicular to the laser beam direction was measured three times at the same height. Figure 9 shows the hardness distribution of the coating from the top to the substrate. The maximum hardness appears near the coating's top, at about 0.1 mm. Combined with the coating's microstructure, as shown in Figure 5, the surface structure of the coating consists of dense equiaxed crystals, while the middle and bonding regions are composed of coarse dendrites. According to the Hall-Petch formula, as the grain size becomes finer and denser, the grain strength would be higher [36,37]. The strength of the metal is negatively correlated with the grain size. Therefore, the strength of the metal is greater and the hardness of the material is higher when the grain size is smaller. The maximum hardness of the coating is 770.7 $HV_{0.5}$. That is about 3.3 times the hardness of the substrate.

$$\sigma_y = \sigma_0 + kd^{-1/2} \tag{4}$$

$\sigma_y$—yield limit of the materials;
$\sigma_0$—lattice frictional resistance when moving individual dislocations;

$k$—related constants;
$d$—average grain diameter.

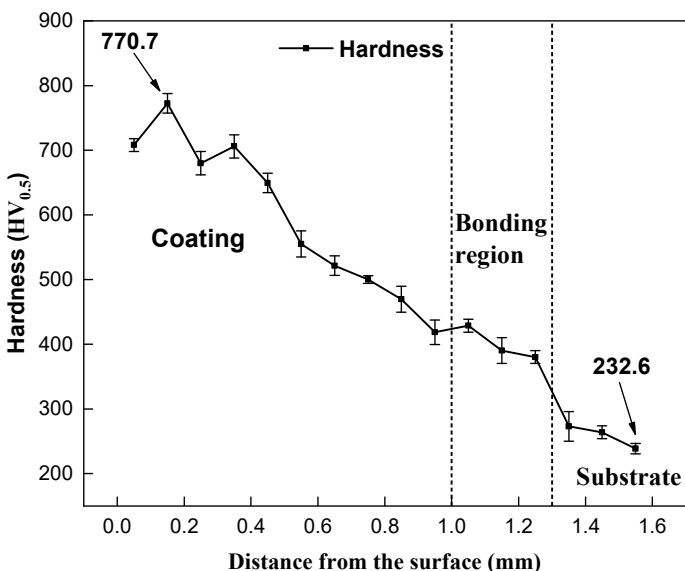

**Figure 9.** Hardness of the coating's cross-section, which is perpendicular to the laser beam direction.

Combined with the X-ray diffraction pattern analysis, the presence of hard phases such as WC, $Fe_3W_3C$, and $Cr_{23}C_6$ can significantly enhance the hardness of the Ni–WC–TiC coating, which is higher than that of the Cr12MoV substrate. However, the hardness decreases from the coating's top to the bottom because there are some substrates at the bottom of the molten pool. The Cr12MoV melts with the high laser energy, and some elements are easy to enter into the molten pool and stay in the coating after cooling. So it has a decreasing trend in the coating. In the bonging region, the hardness also has a decreasing trend and gradually approaches the substrate's hardness because more elements have entered this region. In the substrate region, the hardness will have a small change and be uniform.

*3.4. Wear Properties and Mechanisms*

The wear volume could be calculated based on the length and width of the wear scar when the experiment was conducted at room temperature [38]. Figure 10 shows the frictional wear curve for the Cr12MoV substrate and Ni–WC–TiC coating. The friction coefficient of the Cr12MoV substrate increased at the beginning of friction and then stabilized at around 0.5. The variation of the friction coefficient of the Ni–WC–TiC coating had a short fluctuation state, which increased to the maximum value (0.3) at 360 s and then decreased to 0.2 in the stable stage. The friction coefficient of the Ni–WC–TiC coating was lower than that of the Cr12MoV substrate at all stages.

Figure 11 shows the wear volume of the Cr12MoV substrate and Ni–WC–TiC coating at 30 min. It can be seen from the figure that the wear volumes of the Cr12MoV substrate and Ni–WC–TiC coating are 3.02 $mm^3$ and 0.26 $mm^3$, respectively. The wear volume of the coating is only 8.6% of the substrate, which demonstrates that the Ni–WC–TiC coating has a smaller wear volume and is helpful in decreasing the wear volume of the Cr12MoV substrate. It may have better wear resistance than the Cr12MoV substrate. The improvement in wear resistance is caused by the reinforced phases, such as $\gamma\sim$(Fe, Ni), WC, $Cr_{23}C_6$, $Cr_2Ti$, and TiC.

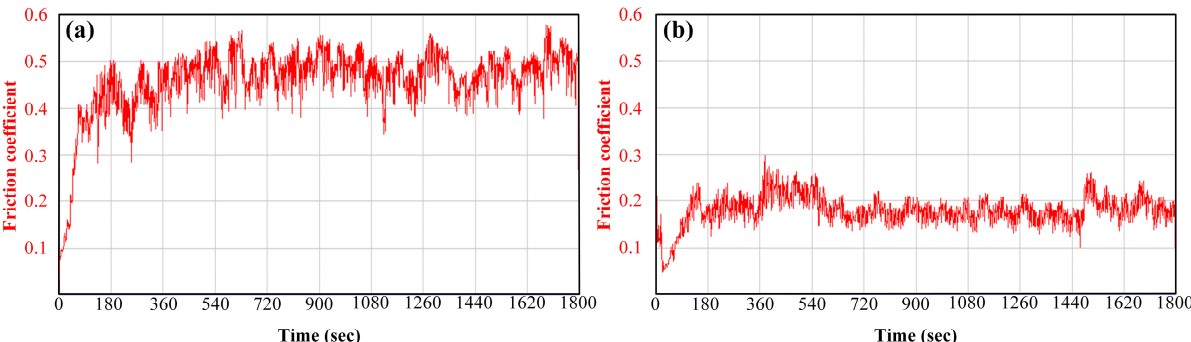

**Figure 10.** Frictional wear curve for the Cr12MoV substrate (**a**) and the Ni–WC–TiC coating (**b**).

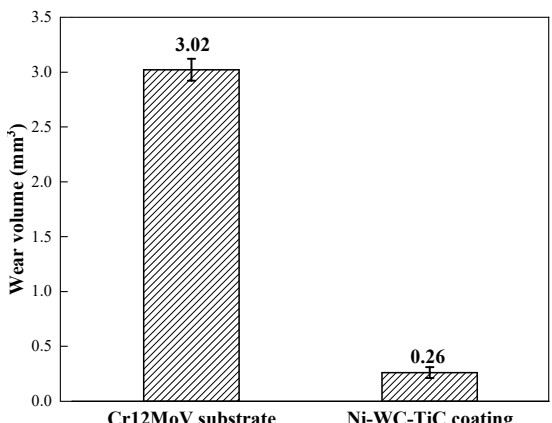

**Figure 11.** The Cr12MoV substrate and Ni–WC–TiC coating's wear volume at 30 min.

To evaluate the long-term wear resistance of the coating, continuous friction wear tests were conducted for 20, 40, and 60 min. Figure 12 shows the wear volume of the Cr12MoV substrate and Ni–WC–TiC coating. It can be seen in the line chart that the wear volumes of substrate and coating increase along with the increase in time. The wear volumes of the Cr12MoV substrate are 1.8 and 4.5 mm$^3$ at 20 and 60 min, respectively. While the wear volumes of the Ni–WC–TiC coating are 0.2 and 0.7 mm$^3$ at 20 and 60 min, respectively. The increase in the amplitude of the coating's wear volume is smaller than that of the substrate. Therefore, the coating may be guaranteed to have good wear resistance during the long period of service.

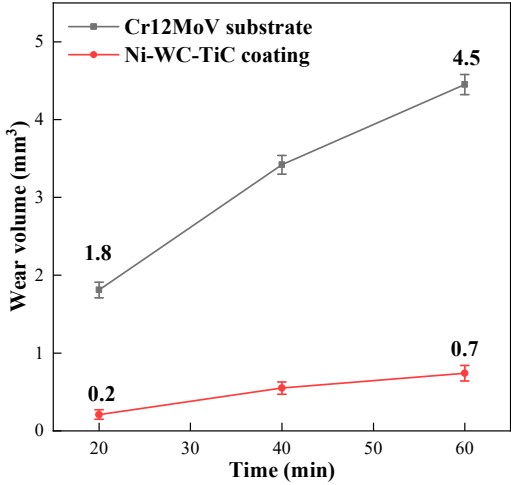

**Figure 12.** Wear volume of the Cr12MoV substrate and Ni–WC–TiC coating at 20, 40, and 60 min.

Figure 13 shows the microscopic morphology of the friction and wear surfaces of the Cr12MoV substrate and Ni–WC–TiC coating. As can be seen in Figure 13a, the worn surface of the substrate is very rough. It has severe plastic deformation, deep pits, and severe spalling. After a certain number of wear cycles, the flakes peeled off from the substrate to form wear debris and adhered to the substrate surface under the action of the normal force of the friction pair. The substrate wear mechanisms are abrasive wear and adhesive wear. In contrast, Figure 13b shows that the furrows on the worn surface of the coating are shallow and narrow, and no significant flaking occurs. This is because the presence of the hard phase helps to resist the plowing effect under the combined action of the friction pair and wear debris during the wear process. The hard phase greatly improves the hardness of the coating, while the resistance to deformation of the coating is improved. The reinforced phases, such as WC, TiC, and $Cr_{23}C_6$, improve the strength of the coating and bring it closer together. However, there is still some wear debris in areas without the hard phase. The substrate wear mechanisms are also abrasive wear and adhesive wear, but there is no brittle spalling.

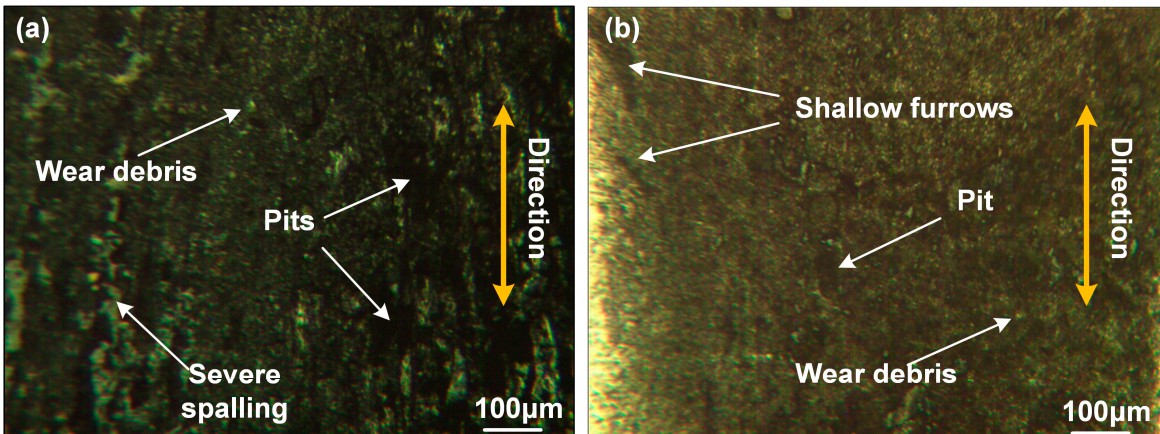

**Figure 13.** Microscopic morphology of friction and wear surface; (**a**) Cr12MoV substrate; (**b**) Ni–WC–TiC coating.

According to the above experimental analysis, the Ni–WC–TiC coating used in this study can effectively improve the hardness and wear resistance of the Cr12MoV substrate, which can extend the service life of the Cr12MoV substrate.

## 4. Conclusions

(1) The Ni–WC–TiC coating was fabricated on the substrate of Cr12MoV die steel by laser. There is a good metallurgical bond between the coating and substrate. The surface morphology of the coating is smooth and uniform;

(2) The coating's microstructures, from top to bottom, are mainly equiaxed crystal, columnar, and cellular dendrite. The phases of the coating are mainly composed of $\gamma{\sim}$(Fe, Ni), $Cr_{23}C_6$, WC, TiC, $Cr_2Ti$, and $Fe_3W_3C$. The grain boundaries are enriched in Cr and C elements, which include some Cr carbides. There are some W and Ti carbides in grains;

(3) The coating's maximum hardness is 770.7 $HV_{0.5}$, which is about 3.3 times the hardness of the substrate. The hardness of the coating surface is higher and gradually decreases down to the substrate. The presence of reinforced phases, such as WC, $Fe_3W_3C$, and $Cr_{23}C_6$, can significantly enhance the hardness of the Ni–WC–TiC coating;

(4) The coating wear volume is 0.26 $mm^3$, which is only 8.6% of the substrate. The friction coefficients of the Cr12MoV substrate and Ni–WC–TiC coating are stable at 0.5 and 0.2, respectively. The friction coefficient of the Ni–WC–TiC coating was lower than that of the Cr12MoV substrate at all stages. This demonstrates that the coating has an important role in improving the wear properties of the substrate.

The reinforced phases, such as WC, TiC, and $Cr_{23}C_6$, improve the strength of the coating and bring it closer together. The wear mechanisms of both the coating and the substrate are abrasive wear and adhesive wear, but there is no significant brittle spalling of the coating.

**Author Contributions:** Conceptualization, Y.L.; Methodology, G.L. and M.Y.; Software, Z.L.; Data curation, Z.L. and G.L.; Writing—original draft, Z.L.; Writing—review & editing, Y.L. and F.D.; Supervision, Y.L. and M.Y.; Project administration, F.D.; Funding acquisition, Y.L. and F.D. All authors have read and agreed to the published version of the manuscript.

**Funding:** This research was funded by the National Natural Science Foundation of China, grant number 51704073; Science and Technology Development of Jilin Province, grant number 20230101335JC/20180520065JH; "13th Five-Year Plan" Science and Technology Research Project of Jilin Provincial Education Department, grant number JJKH20180419KJ/JJKH20180427KJ; Technology Innovation Development Project of Jilin City, grant number 20166013; Central Universities, grant number 3132017009; China Postdoctoral Science Foundation, grant number 2017M611209; Natural Science Foundation of Liaoning Province, grant number 20170540083.

**Data Availability Statement:** Data is only available upon request due to restrictions such as privacy or ethical considerations. The data presented in this study are available on request from the corresponding author.

**Acknowledgments:** This thesis work is supported by Northeast Electric Power University.

**Conflicts of Interest:** The authors declare no conflict of interest.

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
