# Peer review of "Microstructure and Wear Resistance of Ni–WC–TiC Alloy Coating Fabricated by Laser"

_lubricants, doi:10.3390/lubricants11040170_

Round 1

Reviewer 1 Report

The manuscript entitled “lubricants-2255432-Laser” dealing with the application of laser has been reviewed. The paper has been nicely written but needs significant improvement. Please follow my comments.

1.     Add some quantitative results to the abstract.

2.     What is the main research question for this research work? You need to highlight it.

3.     Why substrated was selected Cr12MoV? Please add more note about it.

4.     What is the future direction of this work?

5.     Nickel has many usage in manufacturing method which can be highlighted in your paper. Please read the following article and add to the introduction to show the experimental application of nickel in sandwich structure printing “Sandwich structure printing of Ti-Ni-Ti by directed energy deposition”.

6.     Figure 4 X-rad Difraction” needs more explanation in the text.

7.     Laser has many advantages over the conventional manufacturing method which can be highlighted in your paper. Please read the following manuscript and add it to the literature to show how laser is comparable with the conventional manufacturing.

Laser subtractive and laser powder bed fusion of metals: review of process and production features

Reviewer 2 Report

This paper studies the Ni-WC-TiC coating on the substrate of Cr12MoV die steel. The experimental design is reasonable, though some key details are missing and needed to clarify. I would recommend the publication of this paper after revision. See the comments below:

1. What’s the grain size of different regions of the coating?

2. Figure 5: How were points D and E chosen respectively, and based on what criteria? 

3. For the hardness tests, was the external force applied in the direction of, or perpendicular to the laser beam direction? Please be clear in the experimental description. If it is along the laser beam direction, then Figure 9 is not providing much insights because the decrease of hardness along depth from the surface is well-known as the Nix-Gao model for the indenter and surface effects. (Conclusion #3)

4. For conclusion #1 - By simply looking at large-scale SEM images, I wouldn’t conclude the perfect uniformity of the coating interface that is free of defects. It needs very careful characterization such as TEMs to examine the smaller features to reach that conclusion.

Reviewer 3 Report

The work carried out by the author is in the interest of the research community. But it is required to incorporate the suggestions/comments in their manuscript for further improvement.

1. The abstract should be more quantitative in terms of improvement. Further, the line in the abstract "The main phases are γ~(Fe, Ni), Cr23C6, WC, TiC, Fe3W3C and Cr2Ti" is not clear. rewrite it. Moreover, at least one line about the application should be added to the abstract.

2. The author should write the objective along with the research gap clearly for a better understanding of the reader. 

3. Section 3.1 will be more appropriate after section 2.2.

4. Resolution of Fig 4 Fig 6 Fig 9 and Fig 12 should be increased.

5. The author can find out a better explanation of microstructure analysis at https://doi.org/10.1016/j.mseb.2020.114799

6. The author has performed the wear analysis but they have not reported the tribometer in the experiment section. It must include. what load, what speed, and what type of test was conducted by the author.

7. What type of continuous friction wear tests were performed by the author?

8. The author has not mentioned the repeatability of the test. it creates doubt. Justify it. If did then add the SD in each graph.

9. The author should mention the direction of the test in Fig 12. Also mentioned the friction coefficient data.

10. Add the friction coefficient of the coating in the conclusion. Further, it should also be more quantitative.

Round 2

Reviewer 1 Report

The paper is ready to publish.

Reviewer 3 Report

No Comments